

# Wintertime aerosol chemistry and haze evolution in an extremely polluted city of North China Plain: significant contribution from coal and biomass combustions

Haiyan Li[1,2], Qi Zhang[2], Qiang Zhang[3,4], Chunrong Chen[3], Litao Wang[5], Zhe Wei[5], Shan Zhou[2], Caroline Parworth[2], Bo Zheng[1], Francesco Canonaco[6], André S. H. Prévôt[6], Ping Chen[7], Hongliang Zhang[7], Kebin He[1,4,8]

[1] State Key Joint Laboratory of Environment Simulation and Pollution Control, School of Environment, Tsinghua University, Beijing 100084, China
[2] Department of Environmental Toxicology, University of California, Davis, CA 95616, USA
[3] Ministry of Education Key Laboratory for Earth System Modeling, Center for Earth System Science, Tsinghua University, Beijing 100084, China
[4] Collaborative Innovation Center for Regional Environmental Quality, Beijing 100084, China
[5] Department of Environmental Engineering, Hebei University of Engineering, Handan, Hebei 056038, China
[6] Laboratory of Atmospheric Chemistry, Paul Scherrer Institute, 5232 PSI Villigen, Switzerland
[7] Handix LLC, Boulder, CO 8031, USA
[8] State Environmental Protection Key Laboratory of Sources and Control of Air Pollution Complex, Tsinghua University, Beijing 100084, China

*Correspondence to:* Qiang Zhang (qiangzhang@tsinghua.edu.cn), Kebin He (hekb@tsinghua.edu.cn)

**Abstract.** The North China Plain (NCP) frequently encountered heavy haze pollution in recent years, particularly during wintertime. In 2015-2016 winter, the NCP region suffered several extremely severe haze episodes with air pollution red alerts issued in many cities. In this work, we investigated the sources and aerosol evolution processes of the severe pollution episodes in Handan, a typical industrialized city in the NCP region, using real-time measurements from an intensive field campaign during the winter of 2015-2016. The average ($\pm 1\sigma$) concentration of submicron aerosol ($PM_1$) during December 3, 2015 – February 5, 2016 was 187.6 ($\pm$ 137.5) $\mu g\ m^{-3}$, with the hourly maximum reaching 700.8 $\mu g\ m^{-3}$. Organic was the most abundant component, on average accounting for 45% of total $PM_1$ mass, followed by sulfate (15%), nitrate (14%), ammonium (12%), chloride (9%) and BC (5%). Positive matrix factorization (PMF) with multi-linear engine (ME-2) identified four major organic aerosol (OA) sources, including traffic emissions represented by a hydrocarbon-like OA (HOA, 7% of total OA), industrial and residential burning of coal represented by a coal combustion OA (CCOA, 29% of total OA), open and domestic combustion of wood and crop residuals represented by a biomass burning OA (BBOA, 25% of total OA), and formation of secondary OA (SOA) in the atmosphere represented by an oxygenated OA (OOA, 39% of total OA). Emissions of primary OA (POA), which together accounted for 61% of total OA and 27% of $PM_1$, are a major cause of air pollution in this region during the winter. Our analysis further uncovered that, primary emissions from coal combustion and biomass burning together with secondary formation of sulfate (mainly from $SO_2$ emitted by coal combustion) are important driving factors for haze evolution. However, the bulk composition of $PM_1$ showed comparatively small variations between less polluted periods (daily $PM_{2.5} \leq 75\ \mu g\ m^{-3}$)



and severely polluted periods (daily $PM_{2.5} > 75$ μg m$^{-3}$), indicating relatively synchronous increases of all aerosol species during haze formation. The case study of a severe haze episode, which lasted 8 days starting with a steady build-up of aerosol pollution followed by a persistently high level of $PM_1$ (326.7 – 700.8 μg m$^{-3}$), revealed the significant influences of stagnant meteorological conditions on acerbating air pollution problems in the Handan region. The haze episode ended with a shift of wind which brought in cleaner air masses from the northwest of Handan and gradually reduced $PM_1$ concentration to < 50 μg

m$^{-3}$ after 12 hours. Furthermore, aqueous-phase reactions under higher relative humidity (RH) were found to significantly promote the production of secondary inorganic species (especially sulfate), but showed little influence on SOA.

## 1 Introduction

Atmospheric particles are a complex mixture of multiple species emitted directly to the atmosphere or formed via gas-to-particle conversions. Aerosols have adverse effects on human health (Pope and Dockery, 2006) and may cause climate change

(Pöschl, 2005; Seinfeld and Pandis, 2012), all of which are intrinsically linked to the chemical composition of aerosols. Therefore, it is crucial to gain a quantitative understanding of aerosol composition and evolution processes for accurately assessing the environmental effects of aerosols.

With the rapid economy growth and urbanization in North China Plain (NCP), air pollution in this region becomes a severe problem and has raised global concern in recent years. Hebei Province, located in the NCP region, is known for

persistent air quality problem and extreme haze pollution events. According to the Ministry of Environmental Protection (MEP) of China, 7 out of the top 10 polluted cities in China in 2015 were located in Hebei province. During the extremely severe haze event that occurred in the winter of 2015-2016 in the NCP region, the hourly peak $PM_{2.5}$ concentration in southern Hebei even exceeded 1000 μg/m$^3$, adversely affecting human health. It is well known that the severe air pollution in the NCP region was caused by large anthropogenic emissions and unfavorable meteorological conditions. Emissions of primary $PM_{2.5}$, sulfur

dioxide ($SO_2$), and nitrogen oxides ($NO_x$) from Hebei in 2015 are estimated to account for 8 %, 6 %, and 7 % of China's national total emissions respectively (http://meicmodel.org/), with large contribution from coal and biomass combustions.

Large anthropogenic emissions in the NCP region have downgraded regional air quality significantly. Extensive studies have been conducted to explore the sources and evolution processes of haze episodes in Beijing, especially with the wide application of Aerodyne Aerosol Mass Spectrometer (AMS)/Aerosol Chemical Speciation Monitor (ACSM) for online

measurements of aerosol chemical composition in recent years (Takegawa et al., 2009; Sun et al., 2010; Sun et al., 2012, 2013a, 2013b, 2014, 2015, 2016a, 2016b; Zhang et al., 2014; Hu et al., 2016). These studies have noted that regional air transport from the south or east surrounding regions, unfavorable synoptic conditions, and heterogeneous secondary reactions associated with high RH initiated the rapid formation and persistent evolution of haze episodes in Beijing. During a record-breaking haze episode in wintertime in Beijing, Sun et al. (2014) estimated that regional transport contributed up to 66% to the steep rise of

air pollutants in Beijing. In addition, organic aerosol (OA) was found to be a major component of aerosol particles, accounting for more than one-third of total $PM_1$ mass. The primary OA (POA) from traffic, cooking, biomass burning, coal combustion,



etc., and secondary OA (SOA) have been distinguished and quantified mainly using positive matrix factorization (PMF; Paatero and Tapper, 1994). Recently, a novel PMF procedure, with the multi-linear engine (ME-2) algorithm, was developed to apportion the OA sources in Beijing and Xi'an, allowing for a more objective selection of source apportionment solution (Elser et al., 2016). However, our knowledge on the sources and aerosol evolution processes for the whole region still remains incomplete and is especially limited for areas outside of Beijing. For other cities in the NCP region, such as Hebei province, only a limited number of aerosol studies have been conducted using offline filter-based measurement techniques so far (Zhao et al., 2013; Wei et al., 2014). Due to low time resolution varying from one day to several days, these studies provided relatively limited information on aerosol emission sources and formation processes, thus it remains unclear how the rapid haze evolution happens and what the driving sources are for the air pollution problems in Hebei. Therefore, it is crucial to conduct research in the areas outside of Beijing, especially many provinces subjected to high anthropogenic emissions, which may provide critical information to help air pollution policy making to be more direct and efficient.

In order to fill this knowledge gap, an intensive field campaign with multiple state-of-the-art research instruments was conducted in Handan, a major city in southern Hebei, during the winter of 2015-2016. Handan is located in the intersectional area of four provinces, Hebei, Shanxi, Henan and Shandong, all of which are heavily urbanized and industrialized (Fig. 1a). Handan itself is also well known for heavy industrial outputs of steel, iron and cement, which result in high local emissions of air pollutants. According to the routine monitoring of the China National Environmental Monitoring Center (CNEMC) from 2013 to 2015, Handan is always listed as one of the top 10 polluted cities in China. Hence, this location and its specific conditions allow for a detailed exploration of aerosol chemistry and haze evolution processes under high anthropogenic emissions.

Here, we provide both overview and evolution cycle analyses of aerosol characteristics using aerosol data acquired with an ACSM and collocated measurements of black carbon (BC), meteorological conditions and gas phase species. The sources of OA are investigated in detail using PMF solved with the ME-2 algorithm (Paatero, 1999). The comparison of species diurnal cycles between weekdays and weekends, polluted and non-polluted days, and the variation of aerosol characteristics with increasing $PM_1$ concentration, help us to gain insights into the driving factors for haze evolution. We also examine the impacts of meteorological conditions based on an intense evolution case of submicron aerosol.

## 2 Experimental methods

### 2.1 Sampling site and instrumentation

In situ measurements were conducted at Hebei University of Engineering (36.57°N, 114.50°E) in Handan from December 3, 2015 to February 5, 2016. Our sampling site is situated in the southeast edge of urban Handan, on the roof of a four-story building (~12 m high), surrounded by the school and residential area, ~300 m north of the South Ring Road, and ~400 m northeast of Handa Highway (S313). In this study, the ambient temperature varied from -12.7 to 14.4 ℃, with an average of





1.8 ℃. The prevailing wind came from the northeast and southwest, characterized by low wind speeds (Fig. 1b).

The mass concentrations of non-refractory submicron aerosol (NR-PM$_1$), including organics, sulfate, nitrate, ammonium,
and chloride, were measured in situ using an Aerodyne ACSM. The detailed description of this instrument can be found in Ng
et al. (2011a). In brief, ambient air was sampled through a PM$_{2.5}$ cyclone to remove coarse particles with diameters exceeding
2.5 μm and then traversed a 2-m-long, ½-inch (outer diameter) stainless steel tube at a flow rate of 3 L min$^{-1}$ using an external
pump. A Nafion dryer was installed before the ACSM to dry aerosol samples and maintain the RH below 30%. Subsequently,
only a subset of the flow at ~85 cc min$^{-1}$ was sampled through a 100 μm critical orifice, focusing aerosol particles between 40
nm and 1 μm into the vacuum chamber via an aerodynamic lens. In our study, the ACSM mass spectrometer was operated at
a scanning speed of 200 ms per amu from 10 to 150 amu. By automatically switching 14 cycles between filter mode and
sample mode, the time resolution for the ACSM data in this study was approximately 15 minutes.

Because of the limit of the vaporizer temperature (~600 ℃), the ACSM could not measure refractory species such as BC.
Thus a multi-angle absorption photometer (MAAP, Thermo Scientific model 5012) was deployed in this study for real-time
measurement of BC concentration. The MAAP was operated at an incident light wavelength of 670 nm, with a PM$_1$ cyclone
and a drying system incorporated in front of the sampling line (Petzold and Schönlinner, 2004; Petzold et al., 2005). Online
PM$_{2.5}$ mass concentration was simultaneously measured using a heated Tapered Element Oscillating Microbalance (TEOM
series 1400a, Thermo Scientific). Other collocated instruments included a suite of commercial gas analyzers (Thermo Scientific)
to monitor the variations of gaseous species (i.e., CO, O$_3$, NO, NO$_2$, NO$_x$, and SO$_2$). Meteorological parameters, i.e.
temperature, RH, pressure, wind speed (WS) and wind direction (WS), were obtained by a Lufft WS500-UMB Smart Weather
Sensor. The data reported in this paper are in Beijing Time (BJT: UTC+8).

## 2.2 ACSM Data analysis

The mass concentrations of non-refractory aerosol species and the spectral matrices of OA were processed using ACSM
standard data analysis software (v 1.5.3.5) within Igor Pro version 6.37. The detailed procedures have been described in Ng et
al. (2011a). The default relative ionization efficiency (RIE) values were used for organics (1.4), sulfate (1.19), nitrate (1.1),
and chloride (1.3), whereas the RIE of ammonium (6.28) was directly determined via analyzing pure NH$_4$NO$_3$ particles. To
account for the incomplete detection of aerosol species, a default collection efficiency (CE) value of 0.5 was applied to the
entire data set as aerosol particles were dried before ACSM sampling and the ammonium nitrate fraction was always lower
than 0.4 during the whole period. Although previous studies have shown that aerosol particles may be slightly acidic during
wintertime in the NCP region, particle acidity was not high enough to affect CE values substantially (Sun et al., 2016a). As
shown in Fig. S1 in the supplementary information, the mass concentrations of PM$_1$ (= NR-PM$_1$ + BC) correlated tightly with
total PM$_{2.5}$ mass loadings measured by TEOM (slope=0.88, r$^2$=0.87).

## 2.3 Positive Matrix Factorization of organic aerosol matrix

To determine potential sources of OA, the ACSM mass spectra were processed using the ME-2 algorithm implemented



with the toolkit SoFi (Source Finder) developed by Canonaco et al. (2013). The so-called *a* value approach allows the user to introduce a priori information in forms of known factor profiles or time series to obtain a rather unique solution and thus reduce the rotational ambiguity of the PMF2 algorithm. The spectra and error matrices of organics were prepared according to the protocol summarized by Ulbrich et al. (2009) and Zhang et al. (2011). Given the interferences of the internal standard of naphthalene at $m/z$ 127-129 and the low signal-to-noise ratio of larger ions, we only considered ions up to $m/z$ 120 in this study.

A reference HOA profile, which is an average of multiple ambient data sets taken from Ng et al. (2011b), was introduced to constrain the model performance with *a* value varying from 0 to 1. Following the guidelines presented by Canonaco et al. (2013) and Crippa et al. (2014), an optimal solution involving four factors with *a* value of 0.1 was accepted. Detailed analyses of the factor time series, mass spectra, diurnal patterns, and correlations with external tracers can be found in the supplementary information (Fig. S2-S6). Note that before using the ME-2 engine, we also attempted to perform PMF analysis with the PMF2

algorithm for 1 to 8 factors. The solutions were thoroughly evaluated following the recommendations outlined in Zhang et al. (2011) and the results of three- and four-factor solutions at $f_{peak}$=0 are shown in Fig. S7-S8. The three-factor solution indicates the identification of a coal combustion OA (CCOA), a biomass burning OA (BBOA) and an oxygenated OA (OOA). But the CCOA factor seems to be mixed with the signals from hydrocarbon-like components related to traffic emissions, which is especially evident given the two noticeable peaks in the diurnal profile of the CCOA-factor during morning and evening rush

hours. In the four-factor solution, the additional factor could not be physically explained and showed indications of factor splitting. Solutions with 5 to 8 factors show further splitting and mixing of factors. Our inability to separate an individual HOA factor using the PMF2 algorithm is probably due to the minor contribution of traffic emissions in Handan, consistent with the fact that the PMF2 algorithm tends to have difficulty in accurately retrieving minor factors (Ulbrich et al., 2009).

## 3 Results and discussions

### 3.1 Overview of aerosol characteristics

Frequent and persistent haze episodes were observed during the campaign, especially from December 16 to December 25, 2015, when an extremely polluted and long-lasting haze event occurred. Based on TEOM measurements, only 13 days met the Chinese National Ambient Air Quality Standard Grade II (CNAAQS, 75 $\mu g/m^3$ for the 24 h average of $PM_{2.5}$) for the whole study period of 65 days. In other words, the daily average $PM_{2.5}$ concentrations exceeded the CNAAQS on 80% of the days

(Fig. 2). On December 22, the daily $PM_{2.5}$ concentration reached the highest value of 725.7 $\mu g/m^3$, leading to the first "red" haze alarm ever in Hebei Province. The meteorological conditions were stagnant with calm winds throughout the study period (WS usually less than 1.5 m/s), although relatively high WS (generally > 1.5 m/s) with cleaner air from northwest of Handan occasionally interrupted the haze evolution process (Fig. 2b). The RH varied from 11.7% to 94.8%, generally with higher values for more polluted periods and lower values during cleaner periods. No precipitation occurred throughout the entire

campaign.

Hourly $PM_1$ concentrations fluctuated dramatically from 4.2 $\mu g/m^3$ to 700.8 $\mu g/m^3$ (Fig. 2g). The average $PM_1$



concentration was 187.6 µg/m³, more than twice as high as that observed in the well-known severe haze event that occurred in Beijing in January 2013 (Sun et al., 2014; Zhang et al., 2014). Organics constituted a major fraction of PM₁, contributing 45% on average during this study, followed by sulfate (15%), nitrate (14%), ammonium (12%), chloride (9%) and BC (5%). The

large fraction of organics in PM₁ was similar to previous observations reported in other areas of NCP during wintertime (Sun et al., 2013a; Zhang et al., 2013; Huang et al., 2014). In the daytime, PM₁ was dominated by secondary species because of active photochemistry, whereas the contributions of primary species were significantly increased at night, probably caused by enhanced primary emissions from fuel combustion coupled with shallow boundary layer height (Fig. S9).

Ambient CO is an indicator for the intensities of anthropogenic emissions. The hourly CO concentration was as high as

10 ppm during the study period, higher than those observed in other areas of China (Andreae et al., 2008; Quan et al., 2014; Yang et al., 2015). Interestingly, the temporal pattern of organics tracked well with that of CO (r = 0.84), implying that combustion emissions were a significant source of organic aerosols in Handan. In addition, during severe haze episodes with high NOₓ and CO concentrations, ozone remained at the background concentration of almost zero for several days instead of showing a regular diurnal variation, indicating active ozone titration by NO and a strong influence of primary emissions on

haze pollution in this study.

## 3.2 Source apportionment of organic aerosol

In this study, three POA factors (HOA, BBOA and CCOA) and one SOA factor (OOA) were resolved by analyzing the ACSM OA mass spectra using the ME-2 algorithm. OOA was the largest contributor to OA mass with an average fraction of 39% (Fig. 3). The traffic-related HOA only accounted for 7% of total OA, which was in accordance with the fact that PMF

analysis performed with the PMF2 algorithm had difficulty to retrieve it (see Sect. 2.3 for more details). On average, primary sources dominated the OA mass (61%) during this winter study, consistent with the results from previous winter studies in the NCP region (Sun et al., 2013a; Zhang et al., 2014; Hu et al., 2016; Sun et al., 2016a). The discussion below focuses on the characteristics, sources and processes of each OA factor.

### 3.2.1 Hydrocarbon-like OA

The HOA factor shows a mass spectrum highly similar to those of freshly emitted traffic or other fossil combustion aerosols (Zhang et al., 2005a; Lanz et al., 2007; Li et al., 2016a). Its profile is dominated by alkyl fragment signatures, the $C_nH_{2n+1}^+$ (m/z 29, 43, 57) and $C_nH_{2n-1}^+$ (m/z 27, 41, 55) ion series. The time series of HOA correlated well with those of NOₓ and BC (r = 0.75 and 0.74, respectively, Fig. 3e), two tracers of vehicle emissions. The diurnal pattern of HOA (Fig. 3i) further confirmed the association of HOA with traffic activities, as it showed two obvious peaks during morning and evening rush

hours. On average, HOA only accounted for 7% of total OA in Handan, much smaller than those observed in the nearby megacities of Beijing and Tianjin (Sun et al., 2013a; Wang et al., 2015). The small HOA fraction in this study is consistent with findings from a previous source apportionment study which revealed that transportation was a minor source of atmospheric particles in Handan (Wang et al., 2014). Bivariate polar plots, which present the concentrations of air pollutants



as a function of WS and WD using the OpenAir software (Carslaw and Ropkins, 2012), indicated that HOA was mainly influenced by local emissions sources, in accordance with its primary characteristics (Fig. S10).

### 3.2.2 Coal combustion OA

Although coal combustion has rarely been reported as an important source of organic aerosols in the US or Europe, it is a large emitter of organics in China (Cao et al., 2006). According to Zhang et al. (2008b), organic carbon can contribute up to 70% of emitted $PM_{2.5}$ for different types of coal combustion in China. During wintertime, coal is the primary fuel for various industries (e.g. power generation, steel milling, and cement production) as well as residential heating in the NCP region. Thus a considerable contribution from coal combustion to OA concentration was expected in this study. Compared to HOA and BBOA, the mass spectrum of CCOA showed strong signals at higher $m/z$, especially a significant peak at $m/z$ 115, and the temporal trend of CCOA correlated tightly with that of $m/z$ 115 (r=0.99, Fig. 3). These findings are similar to observations made in Beijing, Changdao, Xi'an, and Lanzhou during winter, where OA factors representing coal combustion were determined (Hu et al., 2013; Elser et al., 2016; Sun et al., 2016a; Xu et al., 2016). Further, a recent study by Zhou et al. (2016) has shown that the ACSM mass spectra of OA from residential coal combustion emissions tend to present a high peak at $m/z$ 115. In addition, CCOA was also found to correlate relatively well with chloride (r = 0.72) during this study, consistent with the fact that coal combustion is also an important emission source of chloride.

Figure 4 compared the OA composition in this study with those of previous winter studies in China. During wintertime, CCOA was observed to contribute a significant fraction of the fine PM mass in regions to the north of the Yangtze River (e.g. Beijing, Lanzhou, and Handan), due to domestic coal combustion for heating in winter. However, little to no CCOA was observed in areas located to the south of the Yangtze River, e.g. Nanjing, Jiaxing, and Ziyang, for which a main reason is the lack of central heating provided by the Chinese government in this region during winter. In this study, similar to the results observed in Beijing and Lanzhou (Sun et al., 2013a; Hu et al., 2016; Xu et al., 2016), CCOA on average accounted for 29% of total OA, with a minimum of 13% at noon and a maximum of 32% at midnight. However, the average mass concentration of CCOA was 23.1 μg/m³ in Handan, much higher than those observed in previous studies. Given the high consumption of coal and the important role of coal combustion for aerosol pollution in Handan, control of air pollutant emissions from coal combustion through technology renewal is essential for air quality improvement in this area.

### 3.2.3 Biomass burning OA

Biomass burning, including wildfires, forest and agricultural burning, and domestic biofuel combustion, is one of the largest emission sources of organics worldwide (Ramanathan et al., 2001). Biomass burning releases air pollutants that have adverse effects on respiratory organs and reduce lung function of human beings (Regalado et al., 2006). In the NCP region, during harvest seasons of summer and autumn, biomass burning tends to largely influence aerosol loadings and characteristics due to open agricultural burning. For example, at a suburban site near Beijing during summertime, Sun et al. (2016b) observed that the contribution of BBOA to OA increased from 6% during the non-biomass burning period to 21% during biomass





burning period. During wintertime, as most previous studies of this region were performed in the megacity of Beijing where coal combustion dominates the energy consumption, BBOA was seldom resolved or found to be a minor fraction of total OA mass (Sun et al., 2013a; Zhang et al., 2014; Huang et al., 2014; Sun et al., 2016a). However, for many small and medium-sized cities in the NCP region, domestic combustion of wood and crop residuals for cooking and home heating is very popular in the countryside during wintertime and could emit large amounts of air pollutants (Zhang et al., 2008a; Ding et al., 2012). For Hebei province, according to the Multi-resolution Emission Inventory for China (MEIC; http://meicmodel.org/), biomass burning contributed 52% to primary organic carbon emissions during the winter of 2015.

In this study, a BBOA factor with high mass concentrations was clearly observed, the mass spectrum of which was characterized by the prominent peaks at $m/z$ 60 and 73, two indicative tracers of biomass burning (Alfarra et al., 2007; Aiken et al., 2009; Lee et al., 2010). The time series of the BBOA varied dramatically and correlated well with that of CO (r = 0.72), which was mainly emitted from combustion-related sources. BBOA showed clear diurnal variations with low mass concentrations occurring during daytime and high mass concentrations arising at night. Consistent with the emission inventory, BBOA on average accounted for 25% of total OA mass, with an average concentration of 20.7 μg/m$^3$, much higher than that observed in other areas of China during wintertime (Fig. 4), indicating the important role of biomass burning emissions in aerosol pollution in Handan. Polar plots showed that high BBOA concentrations were mainly related to local emissions (Fig. S10), probably from biofuel combustions for cooking and residential heating.

### 3.2.4 Oxygenated OA

Although two or more OOA factors with different oxidation degree and formation pathways have been resolved in previous wintertime studies in China (Xu et al., 2015; Sun et al., 2016a), only one OOA factor was observed in this study. The mass spectrum of OOA presented a pattern similar to those reported before (e.g., Zhang et al., 2005b; Ng et al., 2011c) with a prominent peak at $m/z$ 44 (15.8% of the total OOA signal). In addition, OOA showed a temporal trend similar to those of sulfate and nitrate, and correlated strongly with the sum of secondary inorganic species (SIA = sulfate + nitrate + ammonium) (Fig. 5). The polar plots of OOA and secondary inorganic species exhibited similar spatial distributions, with high concentration hotspots located in the northeast, especially during polluted periods (Fig. S10). The temporal variation profile of OOA was much different from those of the POA factors (r$^2$ = 0.25; Fig. 5). As shown in Fig. 3, while POA varied dramatically between day and night due to the influence of local emissions, the mass concentrations of OOA often built up gradually and remained at high levels for several days until being swept away by clean air masses. These results are consistent with OOA being representative of SOA. Although the diurnal profile of OOA was overall flat in this study, the mass fraction of OOA to total OA increased significantly during daytime, reaching a maximum of 64% at 14:00 BJT (Fig. 3n).



### 3.3 Diurnal variations and insights into aerosol sources

**3.3.1 Weekdays versus weekends**

As air pollutants are mainly emitted from anthropogenic sources in Handan, comparing the diurnal profiles of aerosol species between weekdays and weekends would provide insights into the variations of different emission sources and atmospheric processes. Generally speaking, weekdays span Monday to Friday, whereas weekends include Saturday and Sunday. However, because the physical and chemical processes in the atmosphere are not completed instantaneously, the variations of aerosol species may be influenced by the carry-over effect of the previous day. Thus, we alternatively define weekdays from Tuesday to Friday, and weekends only including Sunday. With this classification, differences in the diurnal variations between weekdays and weekends are more visible. Comparisons of the diurnal cycles using the Monday-Friday and Saturday-Sunday definitions are presented in the supplementary information (Fig. S11).

As displayed in Fig. 6, the diurnal variations of meteorological parameters did not significantly change from weekdays to weekends, providing a good opportunity to investigate the influence of anthropogenic activities. As expected, the diurnal pattern of HOA, which is associated with traffic emissions, presented a more distinct morning peak on weekdays. This was also the case for BC, CO, and $NO_x$, which are all fossil fuel combustion tracers. However, the evening rush hour peaks of these species did not show much of a difference between weekdays and weekends, indicating that human activities in the evening were not significantly reduced on weekends. Other aerosol species showed generally similar diurnal trends between weekdays and weekends, similar to the results observed in Beijing (Sun et al., 2013a). In contrast, stronger weekday vs. weekend differences were observed in the US, where the mass concentrations of aerosol species are obviously reduced during weekends (Young et al., 2016; Zhou et al., 2016). Results from this study reveal that active anthropogenic emissions tend to persist throughout the entire week in polluted regions in Handan, leading to limited differences in the concentrations and compositions of major air pollutants between weekdays and weekends. The exception is traffic emissions, for which the morning rush hour peak is more prominent during weekdays.

**3.3.2 Polluted versus non-polluted periods**

To gain further insights into the evolution of aerosol particles throughout the day, especially during hazy conditions, we explored the diurnal differences of meteorological conditions and air pollutants between polluted and non-polluted days (Fig. 7). According to the CNAAQS Grade II of daily $PM_{2.5}$ concentrations (75 μg/m³), only the 13 days (out of a total of 65 days) were found to meet the requirement and are considered to be non-polluted in this study; the rest are defined as polluted periods. Note that of these 13 non-polluted days, only 3 days achieved the 24 h CNAAQS Grade I level of $PM_{2.5}$ (35 μg/m³).

The temperature was relatively low throughout the period, averaging 2.1°C and 0.17°C on polluted and non-polluted days, respectively. The RH during polluted periods was slightly higher during daytime, favoring the aqueous-phase processing of atmospheric pollutants. The influence of RH is discussed in detail in Sect. 3.5. Stagnant weather conditions with lower wind speeds were observed on polluted days, especially during nighttime, which would aggravate the accumulation of aerosol





pollution. Unsurprisingly, the mass concentrations of aerosol components and the mixing ratios of gaseous species were much higher on polluted days. But the diurnal differences between polluted and non-polluted periods could provide some information regarding their evolutionary processes. The diurnal profiles of secondary inorganic species (i.e. sulfate, nitrate, and ammonium), were flatter on polluted days. For example, in the diurnal profile of nitrate during polluted periods, the maximum and minimum concentrations were different by only 13 % or 4.4 $\mu g/m^3$. This behavior is consistent with the comparison of polar plots between polluted and non-polluted days, which indicated a significant effect of regional transport on polluted periods for secondary species. In contrast, the diurnal trends of primary aerosol species, e.g. HOA, BBOA, and CCOA, during polluted periods differed substantially from those during non-polluted periods. Compared to the relatively flat diurnal profiles on non-polluted days, the mass concentrations of HOA, BBOA, and CCOA were strongly enhanced at nighttime on polluted days. This suggests that the sharp increases of primary species at night, especially those of BBOA and CCOA, may play an important role in haze formation.

### 3.4 Evolution of aerosol characteristics with increasing PM$_1$ concentration

Identifying the responsible emission sources and formation pathways during haze events is essential to effectively implement emission controls, especially with the increased frequency of haze events during winter. In this study, the whole period is divided into polluted and non-polluted days, as described in Sect. 3.3.2. The average PM$_1$ concentration during polluted days (211 $\mu g/m^3$) was more than three times higher than that during non-polluted days (49 $\mu g/m^3$). However, the average aerosol composition did not show obvious changes between these two types of days, indicating the synchronous increase of all aerosol species (Fig. 8a). Indeed, during polluted days, the average mass concentrations of all aerosol species, except for BC, were approximately four times as high as those during non-polluted days (Fig. 8b). Sulfate, CCOA, and BBOA showed the highest polluted/non-polluted ratios, which were 5.3, 5.0, and 5.5, respectively (Fig. 8b). Given the higher average RH on polluted days (average±1σ = 56.5±18.8%) compared to non-polluted days (average±1σ = 40.9±18.7%), aqueous-phase processing likely has increased the production of sulfate (Wang et al., 2012; Zheng et al., 2015; Elser et al., 2016). During polluted days, the average oxidation ratio of sulfur (molar ratio of sulfate to sum of sulfate and SO$_2$) was 0.27, higher than that on non-polluted days (0.16). On the other hand, the strong increases of CCOA and BBOA were possibly caused by enhanced gas-to-particle partitioning associated with high PM mass loadings during polluted periods (Mader et al., 2002). Interestingly, compared to aerosol species, CO showed a lower polluted/non-polluted ratio of approximately 2. A possible reason is that CO has a longer atmospheric lifetime compared to aerosol particles, thus it has a more elevated regional background concentration. Note that the polluted/non-polluted ratios for SO$_2$ and NO$_x$ were also lower compared to the aerosol species. This is potentially a result of enhanced aqueous phase oxidation of SO$_2$ and NO$_x$ as well as more efficient wet deposition, since the more polluted periods were generally more humid.

Figure 9 further displays the average hourly variations of the mass fractions of aerosol species as a function of PM$_1$ concentration. Similar to the results observed in Beijing during wintertime, the nitrate fraction in PM$_1$ showed a decreasing trend with increasing PM$_1$ mass loading whereas the contribution of sulfate increased from 12% to 20% as PM$_1$ concentration



increased from 100 µg/m³ to 600 µg/m³. Since it was unlikely that the emission sources of the main gaseous precursors of these two species (i.e., $NO_x$ and $SO_2$) had changed significantly during this study, the observed changes in aerosol compositions suggest different formation mechanisms of nitrate and sulfate during wintertime. The substantially elevated production of sulfate during high PM episodes was likely attributable to higher ambient RH, which facilitated sulfate production through aqueous-phase reactions of $SO_2$ (Kim et al., 2016; Li et al., 2016b; Sun et al., 2013b). The oxidation ratio of sulfur increased

from 0.1 to 0.4 when $PM_1$ concentration raised from ~10 µg/m³ to 600 µg/m³. The mass fractions of different OA factors varied widely as $PM_1$ concentrations increased. The contribution of HOA to total $PM_1$ was minor and remained relatively stable across all mass loadings. However, the mass fractions of CCOA and BBOA increased nearly linearly with $PM_1$ concentrations rising from ~20 µg/m³ to 300 µg/m³ and plateaued at higher aerosol loadings. OOA, a surrogate of SOA, showed the opposite PM-loading dependency, and its contribution decreased slightly with increasing $PM_1$ concentration. The study of Sun et al. (2013a)

in Beijing also found a growing contribution of CCOA and a declining contribution of OOA with increasing $PM_1$ concentrations during wintertime. These results uncovered the important role of POA in the development of high PM pollution during wintertime. Indeed, the scatter plot of OA vs. $PM_1$ concentrations (Fig. 9c) demonstrates that higher mass fractions of organics in $PM_1$ were associated with elevated POA contributions to total OA, especially when $PM_1$ concentrations were more than 200 µg/m³ (Fig. 9c). Overall, the results here suggest that secondary formation of sulfate (mainly from $SO_2$ emitted by

coal combustion), and primary emissions of organics from coal combustion and biomass burning are important driving factors for the development of winter haze pollution in Handan.

**3.5 A case study on an intense haze episode and the influence of meteorological conditions**

     From December 14 to December 28, 2015, an extremely severe haze episode occurred and was characterized with a steady build-up of air pollutants, including fine particles and CO, over a period of ~ 5 days (Dec. 17 -21, 2015) followed by

approximately 4 days of heavy air pollution, during which the average CO mixing ratio was 6.7 ppm and the average $PM_1$ concentration was 500.1 µg/m³ (Fig. 10). This episode ended on Dec. 25, during which stronger winds from the northwest appeared to bring in cleaner air, leading to dramatic reductions of air pollutants. This type of evolutionary process has been frequently observed in Beijing during autumn and winter, and is called "sawtooth cycles" by Jia et al. (2008). In this study, the whole haze cycle was divided into five stages: (1) a clean period (Stage 1), (2) an almost linear increasing period of $PM_1$

concentration (Stage 2), (3) a remarkably high pollution period lasting for four days (Stage 3), (4) an abruptly cleaned up period (Stage 4), and (5) another clean period as the start of a new cycle (Stage 5). As shown in Fig. 10, each stage was initiated by a sudden change in the WD and air masses from different regions via the HYSPLIT back trajectories (Draxler and Rolph, 2013). This indicates that meteorological change is an important driving force during the evolution of haze episodes.

     Stage 1 was characterized with high winds from the northwest, which brought clean air masses from Western Siberia.

Aerosols associated with this air mass origin were largely free of high anthropogenic emissions and appeared to be aged with a high contribution of secondary species. Consistently, the CO concentration during stage 1 was relatively low. During stage 2, the WD changed and the WS was lower than 1 m s⁻¹. The air masses from the northern and southern areas of Handan were





influenced by high anthropogenic emissions in northern Hebei and Henan province, respectively. Thus, the PM$_1$ concentration steadily increased during this stage, with an average of 164.6 μg/m$^3$. Stage 3 was dominated by southerly and northerly winds

and really stagnant conditions with low WS. On December 23, air masses from the southern and northern areas of Hebei circulated around Handan, leading to the accumulation of air pollutants including PM and CO. The average PM$_1$ concentration during stage 3 was 500.1 μg/m$^3$, with the hourly maximum reaching as high as 700.8 μg/m$^3$, much higher than that observed during the severe haze episode in Beijing in January 2013 (~300 μg/m$^3$; Sun et al., 2014). Accompanied with a high CO concentration (average of ~7 ppm) during stage 3, O$_3$ concentration remained at a very low level of almost zero and with

minimal diurnal variations, suggesting that gas-phase oxidation might not be a dominant mechanism for haze formation. Moreover, stage 3 was characterized with high RH, exceeding 70% most of the time, which would promote the aqueous-phase formation of secondary species. Indeed, a high mass fraction of secondary species, especially a notable increase in sulfate contribution, was observed during stage 3. During stage 4, due to the return of cleaner air masses long transported from northwest, air pollutant concentrations in Handan decreased dramatically and PM$_1$ concentration decreased from 443.7 μg/m$^3$

to 34.1 μg/m$^3$ within only 12 hours.

To further evaluate the influence of air mass origins on aerosol characteristics, we performed the cluster analysis of HYSPLIT back trajectories for the whole study period to elucidate the relationship between aerosol concentration or composition and different clusters. As shown in Fig. S12, the whole NCP region was heavily polluted, with high PM$_1$ concentrations for all four clusters. Overall, the aerosol compositions were similar among different clusters. However, we

indeed observed an important role played by winds in altering aerosol characteristics according to the above case study. Referring to the haze cycle analysis, we attempted to apply another classification method based on WD and WS. Periods with WS exceeding 1.5 m s$^{-1}$ from the northwest of Handan were denoted as "NW_HWS", whereas the remaining periods were classified as "Others" (Fig. 11). As expected, the PM$_1$ concentration of "Others" was more than six times higher than that of "NW_HWS". Secondary aerosol species (i.e. sulfate, nitrate, ammonium and OOA) contributed 66% of total PM$_1$ for

"NW_HWS". As air masses associated with "Others" were more strongly influenced by anthropogenic sources, the main primary species (i.e. HOA, BBOA, CCOA and BC), accounted for a higher fraction of 32% for "Others". These results highlight the importance of high winds from the northwest of Handan in alleviating PM levels and changing aerosol composition during wintertime.

As mentioned previously, the sulfate contribution during stage 3 was visibly enhanced under high RH, revealing the

effects of RH on aerosol processing. Many previous studies have observed the increased production of secondary inorganic aerosol species through aqueous-phase processing. In this study, we used the oxidation ratios of sulfur and nitrogen, defined as $f_S = nSO_4^{2-}/(nSO_4^{2-} + nSO_2)$ and $f_N = nNO_3^-/(nNO_3^- + nNO_x)$, respectively, to explore the influence of RH on aerosol formation (Fig. 12). Under relatively dry conditions (RH<50%), both $f_S$ and $f_N$ were almost constant. However, when RH>50%, $f_S$ started to increase linearly, similar to the results observed by Zheng et al. (2015) in Beijing. In comparison,

$f_N$ increased with a much slower speed, suggesting different roles of aqueous-phase reactions in the formation of sulfate and nitrate. Recently, studies of aqueous-phase chemistry have paid increasing attention to organic components. Ge et al. (2012)





observed the strong enhancement of SOA during a fog event in the Central Valley of California during winter. Based on high resolution mass spectra from an AMS, Sun et al. (2016a) retrieved an aqueous-phase-processed SOA (aq-OOA) that tracked well with RH in Beijing during wintertime. However, the mass fraction of SOA in total OA in this study remained relatively stable and showed no dependency on RH (Fig. 12c). The RH-binned bulk composition of submicron aerosol also only exhibited an obvious increase of sulfate at high RH (Fig. S13). One explanation for this observation is that the variations of SOA contribution may be largely interfered by high fractions of POA across different RH values. For another, a portion of OOA formed through aqueous phase reactions may be incorporated into fog droplets, which are too large to be transmitted into the ACSM aerodynamic lens, as reported by Ge et al. (2012). This explanation is consistent with the results obtained by studying a fog event in London, in which no increase in OOA concentration was detected by AMS measurement, whereas the single particle mass spectrometry observed aqueous-phase SOA production (Dall'Osto et al., 2009).

## 4 Conclusions

In order to characterize aerosol sources and formation processes under high anthropogenic emissions in the NCP region, a field campaign was conducted in Handan during the extremely polluted winter of 2015-2016. For the entire study period, only 13 out of 65 days met the Chinese NAAQS Grade II of 75 $\mu g/m^3$ for daily $PM_{2.5}$. The average concentration of submicron aerosol was 187.6 $\mu g/m^3$, with hourly values fluctuating dramatically by a factor of ~ 150, from 4.2 $\mu g/m^3$ to 700.8 $\mu g/m^3$. Organics dominated the bulk composition of submicron aerosols (44.6% of $PM_1$ mass), similar to previous observations in the NCP region during wintertime. PMF analysis identified three primary sources of organic aerosol, i.e. traffic, coal combustion and biomass burning, and one SOA factor. CCOA was the largest contributor to POA, on average accounting for 29%, followed by BBOA (25%). The mass fraction of HOA in total OA was only 7%, indicating the minor contribution of traffic emissions in Handan. Although the aerosol concentration during polluted days was more than three times higher than that during non-polluted days, little variation was observed in the average aerosol bulk composition, revealing the relatively synchronous increase of all aerosol species during haze evolution. Compared to non-polluted days, the stagnant weather conditions, with low wind speed and high RH, and the strong enhancement of primary species at nighttime, prompted haze formation during polluted days. Moreover, the variation of aerosol mass fractions with hourly increasing $PM_1$ concentration further revealed that the secondary formation of sulfate (mainly from $SO_2$ emitted by coal combustion) and primary emissions from coal combustion and biomass burning, are important driving factors of haze formation. This is mainly related to the low efficient combustion of coal and biomass fuels during wintertime. Overall, sulfate, chloride, and CCOA on average accounted for a total of 37% of $PM_1$ mass (Fig. 1c), elucidating the important role of coal combustion in air pollution in Handan. Given the continuing high consumption of coal for various industries and residential heating in winter, technology-based emission controls on coal combustion would effectively improve the air quality in Handan.

A severe haze episode that started with a steady build-up of aerosol pollution followed by an abrupt clean period was studied. Our results indicate the strong influence of meteorological conditions on haze evolution. Due to high anthropogenic





emissions around Handan, the whole study region was heavily polluted through the cluster analysis of back trajectories, with high aerosol concentrations for all clusters. However, high aerosol loadings can be rapidly alleviated by strong winds from the northeastern areas. Under high RH (RH>50%), the oxidation ratio of sulfur increased linearly, suggesting the important role of aqueous phase chemistry in sulfate formation during wintertime. Generally, results in this study may provide useful insights into aerosol chemistry and haze evolution in Hebei province during wintertime, and have important implications for pollution control in this heavily polluted area.

**Acknowledgements**

This work was funded by the National Natural Science Foundation of China (41571130035 and 41625020). Haiyan Li was partially supported by the Doctoral Short-Term Visiting-Abroad Foundation of Tsinghua University, Beijing. Qi Zhang acknowledges the Changjiang Scholars program of the Chinese Ministry of Education. We also give special acknowledgement to lab members in the Department of Environmental Engineering, Hebei University of Engineering, Handan, China, whose
help was invaluable in setting up this field campaign.

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





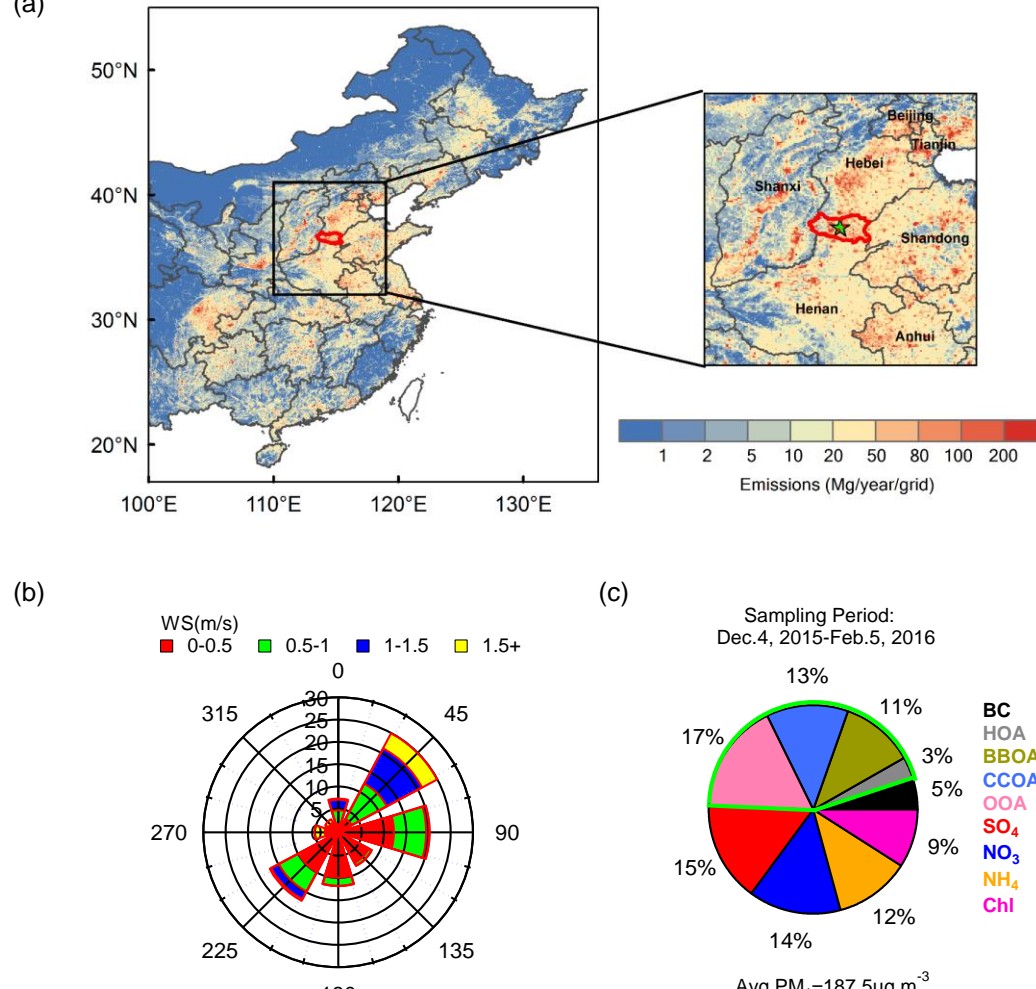

**Figure 1.** (a) Location of the sampling site in Handan in North China Plain. The map is color-coded by annual organic carbon emission rates modeled by Multi-resolution Emission Inventory for China (MEIC, http://www.meicmodel.org). The grid size is 0.05° × 0.05°. (b) Wind rose plot colored by wind speed for the entire period. Radial scales correspond to the frequency. (c) Compositional pie chart of submicron aerosol for the whole study, where the total organic fraction is outlined in green.






**Figure 2.** Time series of (a) ambient air temperature (T) and relative humidity (RH); (b) wind direction (WD) colored by wind speed (WS); (c) mixing ratios of $NO_2$ and $SO_2$; (d) mixing ratios of CO and $O_3$; (e) mass concentrations of organics, sulfate and nitrate; (f) mass concentrations of ammonium, chloride, and black carbon; (g) mass fractional contribution of chemical species to total $PM_1$ with the time series of total $PM_1$ concentration plotted in black on the right y-axis; (h) mass fractional contribution to total OA mass of the four factors derived from PMF analysis with the time series of organic aerosol plotted in green on the right y-axis. Days violating the CNAAQS for $PM_{2.5}$ (= 75 $\mu g\ m^{-3}$) are highlighted in the shade of pale green.





**Figure 3.** (a-d) Mass spectra of hydrocarbon-like OA (HOA), coal combustion OA (CCOA), biomass burning OA (BBOA), and oxygenated OA (OOA). (e-h) Time series of OA factors and the corresponding tracer compounds. (i-l) Diurnal patterns of OA factors. (m) Average fractional pie chart of OA factors to total OA for the campaign. (n) Average diurnal mass contributions of OA factors to total OA, with the average diurnal concentration of organics on the right y-axis.



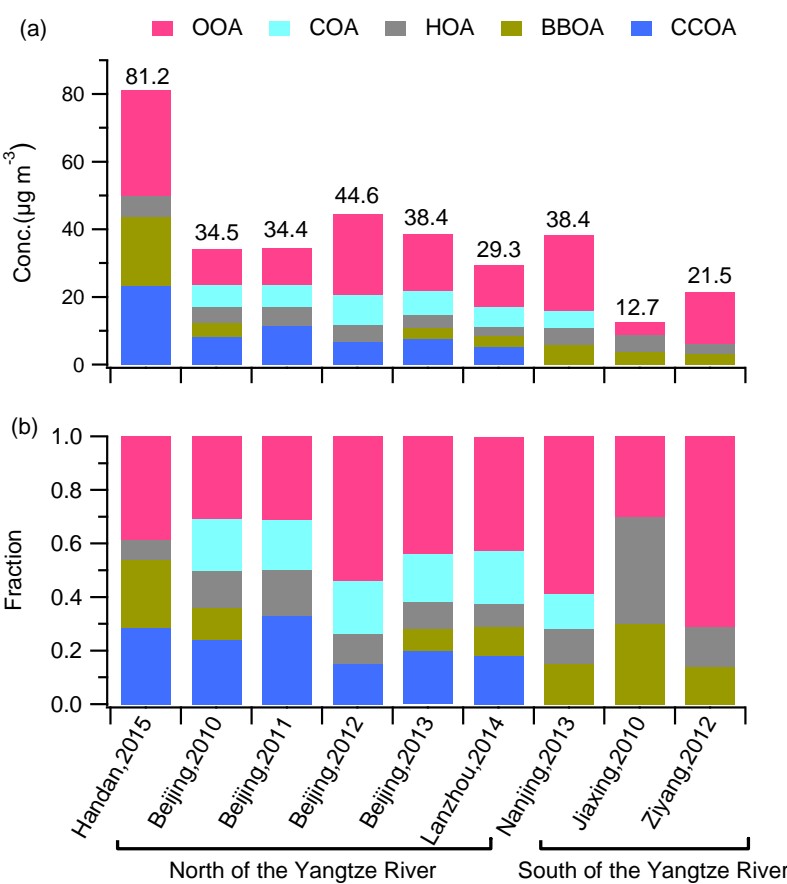

**Figure 4.** Summary of the average (a) mass concentration and (b) chemical composition of organic aerosols from winter studies in China. The total concentration of OA ($\mu g/m^3$) is shown on the top of the bar in panel (a). See Table S1 in the Supplement for detailed information.





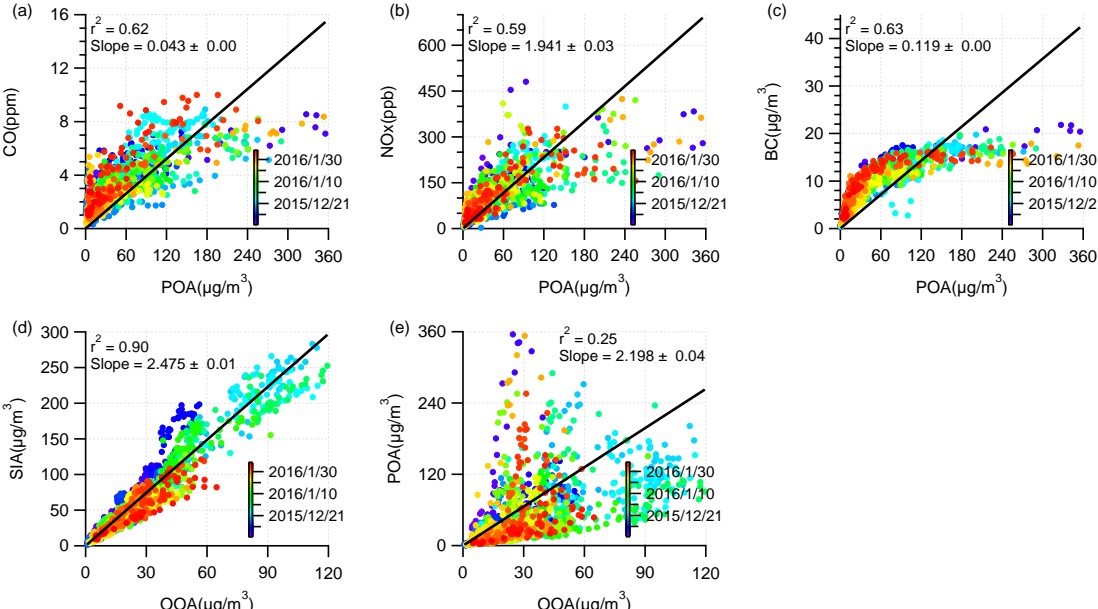

**Figure 5.** Scatter plots of (a) CO vs POA, (b) $NO_x$ vs POA, (c) BC vs POA, (d) SIA vs OOA, and (e) POA vs OOA.

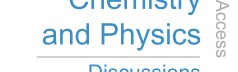



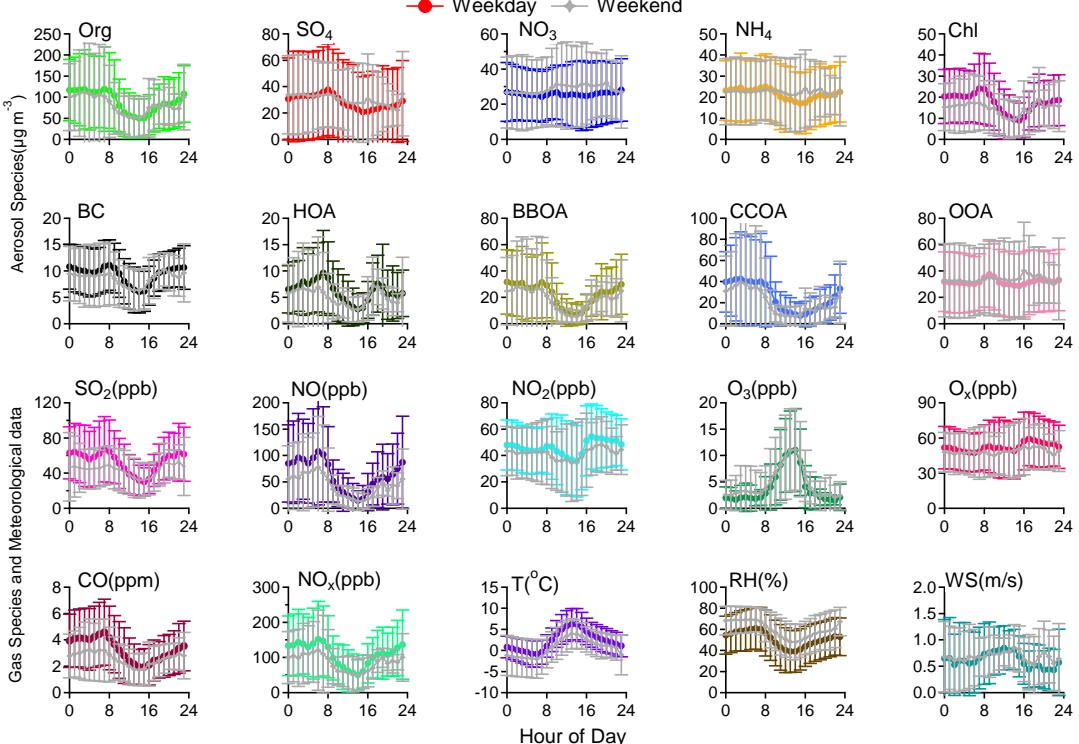

**Figure 6.** Average diurnal profiles along with the standard deviation of PM$_1$ species, four OA factors identified by PMF analysis, various gas-phase species, and meteorological parameters on weekdays (Tuesday to Friday inclusive) and weekends (Sunday only) during the campaign.

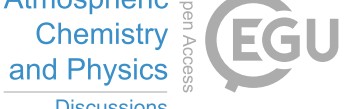



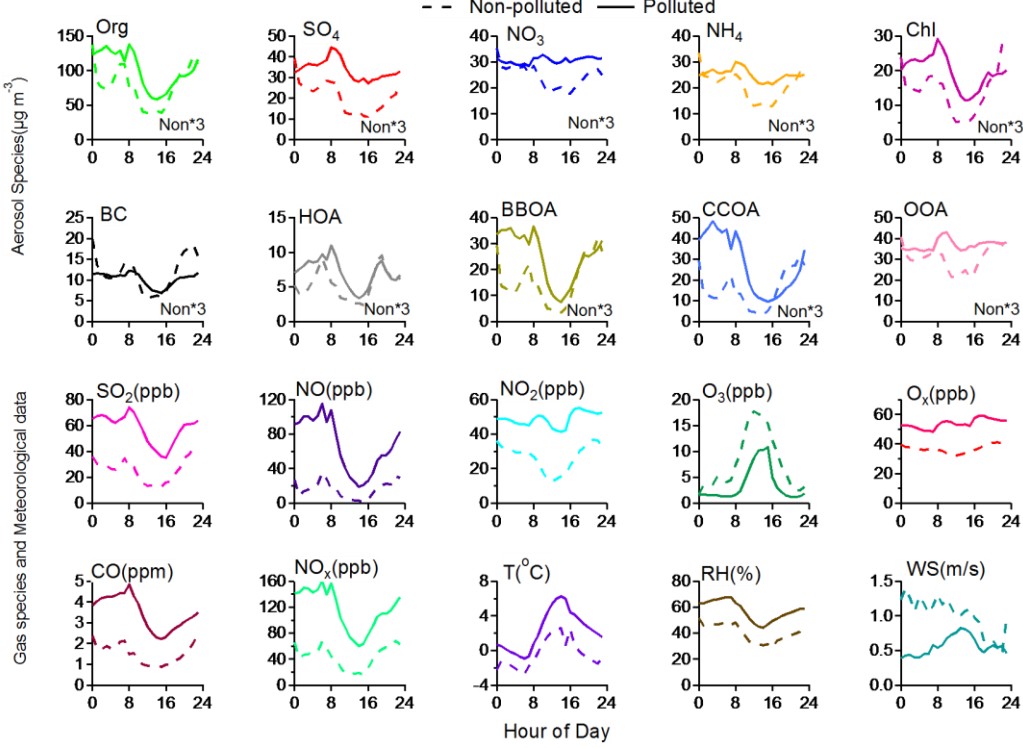

650

**Figure 7.** Average diurnal cycles of PM$_1$ species, four OA factors identified via PMF analysis, various gas-phase species, and meteorological parameters on polluted and non-polluted days. The mass concentrations of aerosol species during non-polluted periods are scaled by three factors in to highlight the differences in their diurnal trends on polluted and non-polluted days.





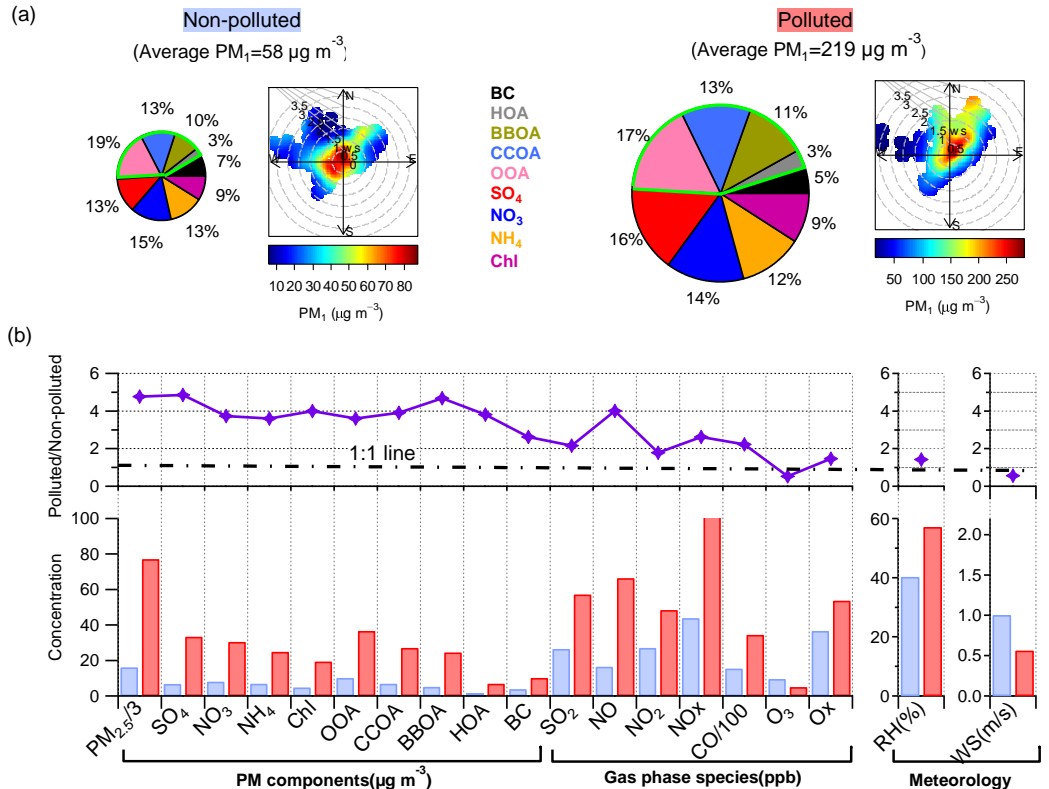

**Figure 8.** (a) Average $PM_1$ composition and bivariate polar plots of $PM_1$ concentration as a function of wind speed and wind direction for polluted and non-polluted periods. (b) Average concentration of PM components, gas phase species and average meteorological conditions during polluted and non-polluted days, with their polluted/non-polluted ratios shown in the top panel.





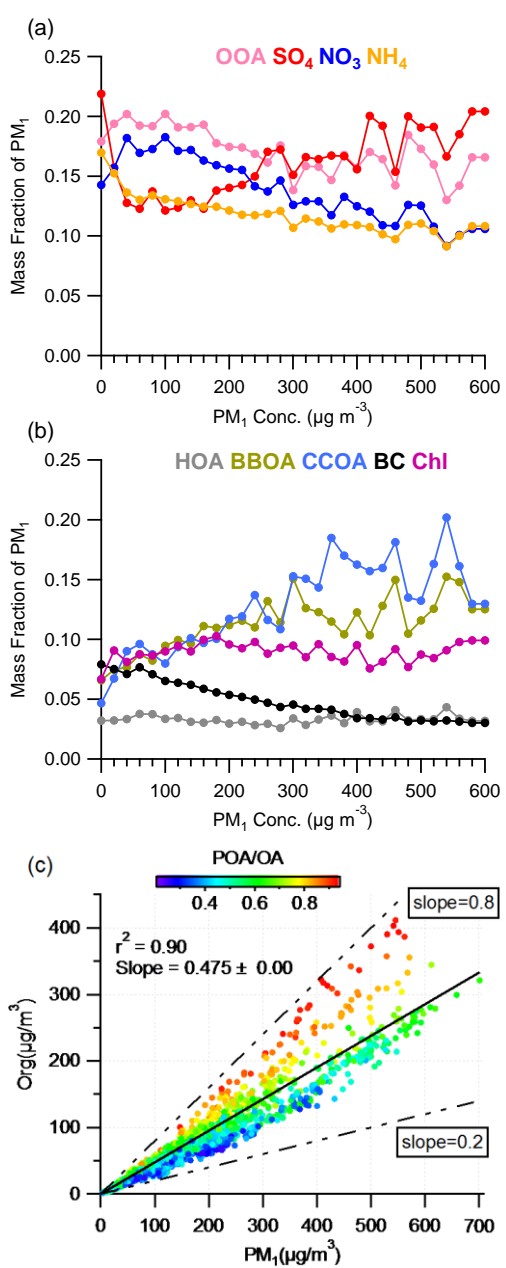

**Figure 9.** (a, b) Variations of the mass fractions of aerosol species as a function of $PM_1$ concentration. (c) Correlation plot of organics and $PM_1$ concentrations, colored by the mass fraction of POA in total OA.



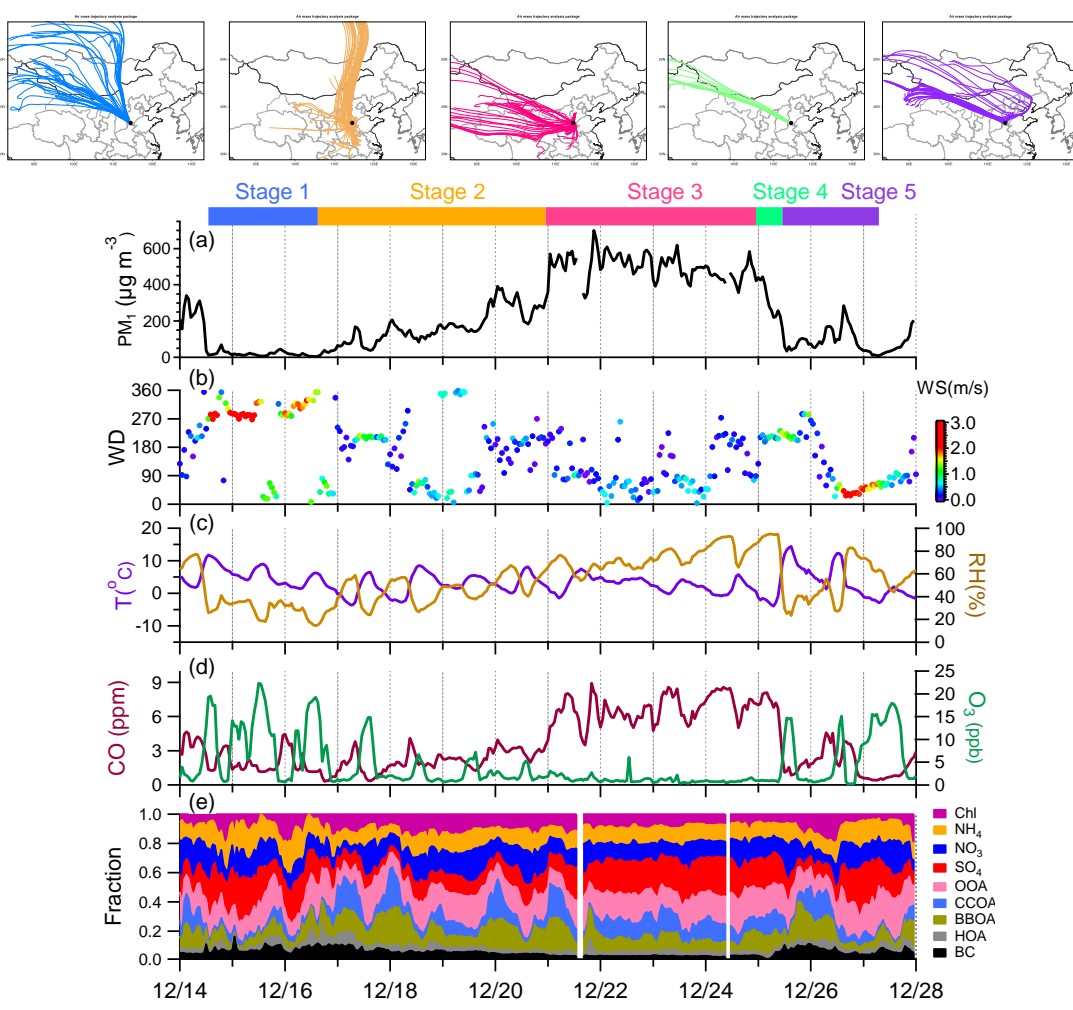

**Figure 10.** Evolution of (a) PM$_1$ concentration; (b) wind direction (WD) and wind speed (WS); (c) Temperature (T) and relative humidity (RH); (d) mixing ratios of CO and O$_3$; (e) mass fractions of aerosol species during a severe haze cycle from December 14 to December 28, 2016. The event was divided into five stages, with back trajectories of each stage shown on the top.

665





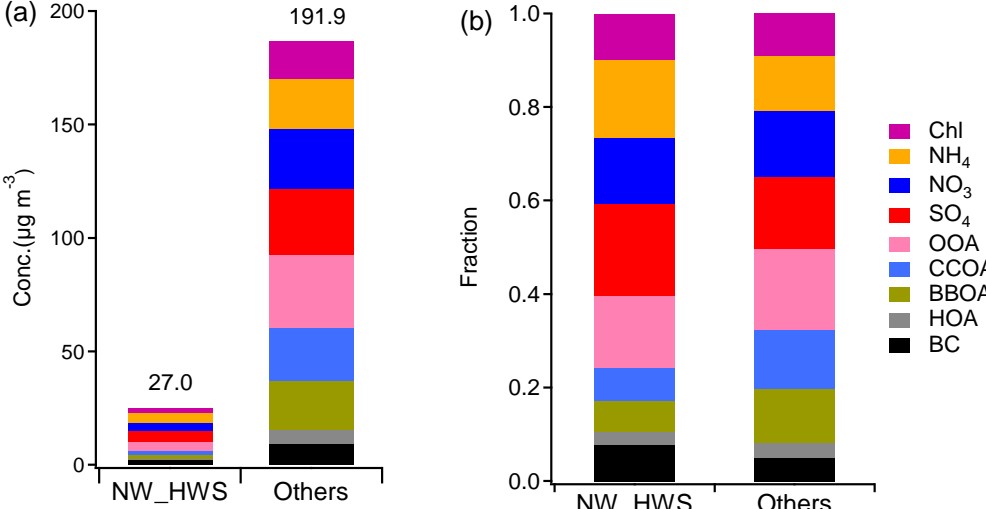

**Figure 11.** Comparisons of (a) mass concentrations of all PM₁ species and (b) fractional contributions of PM₁ species between "NW_HWS" and "Others". The "NW_HWS" refers to high winds from the northwestern areas, and "Others" refers to the remaining.

670





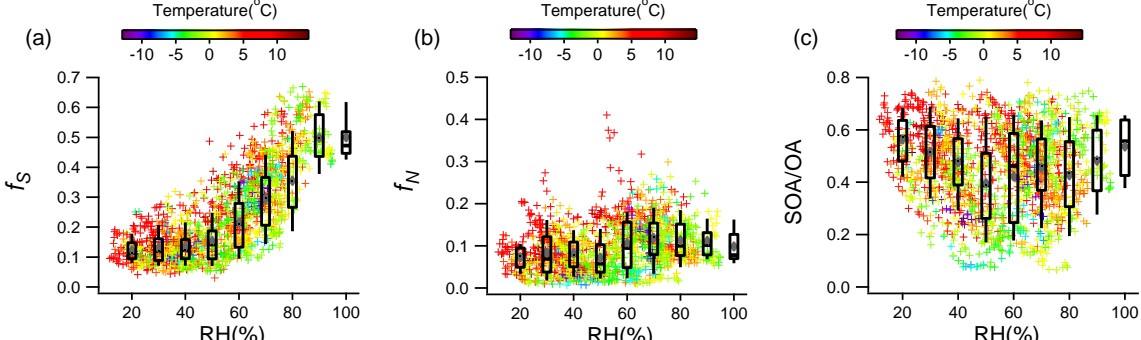

**Figure 12.** Variations of (a) $f_S$, (b) $f_N$, and (c) the mass fraction of SOA in total OA plotted against increasing RH. The data are also binned according to RH values, and the mean (cross), median (horizontal line), 25th and 75th percentiles (lower and upper box), and 10th and 90th percentiles (lower and upper whiskers) are shown for each bin.