# Peer review of "Wintertime aerosol chemistry and haze evolution in an extremely polluted city of North China Plain: significant contribution from coal"

_Atmospheric Chemistry and Physics, 2016_

## Referee Comment (RC1) · Anonymous Referee #2 · 28 Jan 2017

General Comments

This manuscript discusses a wintertime field campaign in the North China Plain during an extreme haze period. The authors evaluate the sources of primary and secondary PM and discuss the evolution of PM constituents and gaseous pollutants in light of prevailing meteorological conditions. The paper is well-written and provides full documentation of the methods, calculations, analyses performed, and conclusions based on these analyses. Given the manuscript's focus on field measurements, the evolution of pollutants within air polluted air masses, and a geographic area that hasn't been the

subject of many intensive field campaigns, this manuscript falls within the scope of ACP. I have no major comments on the manuscript, but present a set of minor comments in the "Specific Comments" and "Technical Corrections" sections below. I do note that for future manuscripts, the authors should ensure that every line is numbered, as reviewers of this manuscript had to count lines to ensure that the correct line numbers were cited.

Specific Comments

1. Line 44: "may cause climate change" is very vague. This should be expanded to a sentence discussing radiative forcings and the indirect and direct effects of PM.

2. Line 153: The CNAAQS is quite high compared to PM standards in other countries. It would be helpful to compare the CNAAQS here to international standards to give readers a broader picture of the percent of days that had high PM.

3. Line 155: Explain how "red haze alarms" are calculated.

4. Figure S9: This figure is a key piece of information related to your argument in lines 170-175. I suggest moving this figure from the supplement to the main document.

5. Line 173: I don't agree with your definition of "background." It would be more correct to simply state that ozone concentrations were nearly zero during haze episodes, as these episodes cannot be considered background time periods.

6. Line 194: Define the acronyms WS and WD here.

7. Line 195: It's not clear how figure S10 supports this argument, and this should be clarified.

8. Lines 216-218: The final sentence in this paragraph is an opinion, not a result, and therefore should be moved to the conclusions.

9. Line 385: What evidence do you have that fN increased? To me, it appears that the median fN at 90-100% is the same as at 20-40%.

10. Lines 412-413: Clarify what is meant by "low efficient combustion."

Technical Corrections

1. Line 56: "combustions" should be "combustion"

2. Line 225: "the" should be inserted between "during" and "biomass"

3. Line 392: "For another" doesn't fit well here. I suggest changing this to "Another explanation is that"

4. Line 421: "northeastern" should be changed to "northwestern"

---

## Referee Comment (RC2) · Anonymous Referee #1 · 29 Jan 2017

This paper presents ACSM results from a heavily polluted city in China during wintertime. The PM1 concentrations averaged at 187.6 ug/m3, in urgent need to elucuiate the characteristics of the PM pollution. The paper is overall well written and the figures are informative, i recommend its publication after addressing a few issues listed below

(1) This reviewer finds that some necessary discussions are lacking in the manuscript. The PM1 contains a significant fraction of chloride - 9%. This fraction is in fact higher than the chloride level typically observed in other AMS studies. Considering the PM1 concentration is high, chloride concentration is also significant. I think its sources,

[Figure]

formation and other charateristics should be discussed. (2)Similar as the comment 1, characteristics of BC should be discussed in more details as well.

(3)Introduction: The authors state that regional transport is a major factor for the heavy haze formation in Beijing. Any comments on the role of local new particle formation and growth?

(4) The authors sometimes use r and sometime r2, it is better to be consistent throughout the manuscript.

(5)The mass fraction of PM1 to TEOM-determined PM2.5 is about 88%, which appeared to be higher than those values in other studies. This probably can be discussed in more details.

(6) The PMF-ME2 algorithm didn't resolve a cooking OA factor. Did the authors try to use any sort of reference COA profile in the PMF analyses, and were any reference BBOA, CCOA profile used as input?(as it was only mentioned HOA profile was used as input).

(7)The weekdays didn't include monday and the weekends didn't include saturday. Was this treatment consistent with and were often used in previous studies?

(8)In Figure 9, this reviewer feels the nitrate fraction increased and then decreased with the increase of PM1 loading rather than only a general decreaing trend.

(9)Figure 11 is for the entire study period, correct? But it was placed in the case study, it probably should be made clear in the figure caption to avoid confusion.

(10)in line 170-175, the authors stated that combustion emissions were significant as it had a good correlation with CO, but the traffic HOA in factor is only a minor contributor to the total OA. I understand the combustion does not only refer to traffic combustion emissions. But this probably should be stated clearly.

---

## Author Comment (AC1) · 7 Mar 2017

**Wintertime aerosol chemistry and haze evolution in an extremely polluted city of North China Plain: significant contribution from coal and biomass combustion**

**By Haiyan Li et al.**

We thank the reviewers for their thoughtful and constructive comments. We have carefully revised the manuscript accordingly. Our point-to-point responses can be found below, with reviewer comments repeated in black and author responses in blue. Changes made to the manuscript are in quotation marks.

**Author Responses to Anonymous Referee #1**

This paper presents ACSM results from a heavily polluted city in China during wintertime. The PM1 concentrations averaged at 187.6 ug/m3, in urgent need to elucidate the characteristics of the PM pollution. The paper is overall well written and the figures are informative, I recommend its publication after addressing a few issues listed below.
(1) This reviewer finds that some necessary discussions are lacking in the manuscript. The PM1 contains a significant fraction of chloride - 9%. This fraction is in fact higher than the chloride level typically observed in other AMS studies. Considering the PM1 concentration is high, chloride concentration is also significant. I think its sources, formation and other characteristics should be discussed. (2) Similar as the comment 1, characteristics of BC should be discussed in more details as well.

Detailed discussions about chloride and BC have been added in Section 3.1. "The average chloride contribution (9%) is relatively high compared to that previously observed in NCP region. Submicron nonrefractory chloride in the aerosol phase can be directly emitted from different sources (e.g., biomass burning and coal combustion) (Lobert et al., 1999; McCulloch et al., 1999), and formed in the atmosphere through gas-to-particle conversion (e.g., $NH_4Cl$ partitioning) (Baek et al., 2006). Considering that chloride demonstrated pronouncedly enhanced peaks during night and it showed good correlations with CCOA and BBOA (r=0.72 and 0.80, respectively), a large fraction of chloride during wintertime was thought to be from primary emissions at night. On average, BC accounted for 5% of total $PM_1$. Its distinct peaks at morning and evening rush hours suggested that BC was mainly associated with traffic emissions."
(3) Introduction: The authors state that regional transport is a major factor for the heavy haze formation in Beijing. Any comments on the role of local new particle formation and growth?

In the introduction, we have added the following information of the importance of new particle formation and growth on haze formation in NCP region: "New particle formation and growth also plays an important role in haze formation. By examining in detail the haze events under typical fall conditions in Beijing, Guo et al. (2014) indicated that nucleation consistently preceded a polluted period with high number concentrations and the development of the episode involved efficient and sustained growth from the nucleation-mode particles over multiple days."
(4) The authors sometimes use r and sometime r2, it is better to be consistent throughout the manuscript.

We have changed all the r2 to r to be consistent throughout the manuscript.
(5) The mass fraction of PM1 to TEOM-determined PM2.5 is about 88%, which appeared to be higher than those values in other studies. This probably can be discussed in more details.

According to the values reported in the literature, the ratio of $PM_1$ to TEOM-determined $PM_{2.5}$ mostly varied from 0.67 to 0.77 in NCP region (Sun et al., 2013; Sun et al., 2014; Zhang et al., 2014; Sun et al., 2015; Hu et al., 2016). But a high $PM_1/PM_{2.5}$ ratio of 0.90 was also observed by

Jiang et al. (2015) during wintertime in Beijing. So the value of 0.88 for $PM_1/PM_{2.5}$ in this study appeared to be a bit higher. The difference may be due to: (1) the contribution of semi-volatile species to $PM_{2.5}$ varied greatly among different periods and different locations, because TEOM is heated to 50 ℃ during the measurement, which might have caused significant losses of semi-volatile species, e.g., ammonium nitrate and semi-volatile organics; and (2) the contribution of particles in the size range of 1-2.5 µm to the total $PM_{2.5}$ might also be different among different pollution episodes and different sites. Detailed discussion has been added in the manuscript: "Compared to the results reported previously in this area (Sun et al., 2013a; Sun et al., 2014; Zhang et al., 2014; Sun et al., 2015; Hu et al., 2016), the ratio of $PM_1$ to TEOM-determined $PM_{2.5}$ in this work appeared to be a bit higher. The difference may be due to: (1) the contribution of semi-volatile species to $PM_{2.5}$ varied greatly among different periods and different locations, because TEOM is heated to 50 ℃ during the measurement, which might have caused significant losses of semi-volatile species, e.g., ammonium nitrate and semi-volatile organics; and (2) the contribution of particles in the size range of 1-2.5 µm to the total $PM_{2.5}$ might also change among different pollution episodes and different sites."

(6) The PMF-ME2 algorithm didn't resolve a cooking OA factor. Did the authors try to use any sort of reference COA profile in the PMF analyses, and were any reference BBOA, CCOA profile used as input? (as it was only mentioned HOA profile was used as input).

We only used the HOA reference profile as a constraint of the PMF-ME2 algorithm in this study. As mentioned in Section 2.3, to do OA source apportionment, we attempted to perform PMF analysis with the PMF2 algorithm at first, which requires no a priori information about factor profiles or time trends. But in the PMF solution, the resolved CCOA factor seemed to be mixed with the signals from HOA, especially considering the two noticeable peaks during morning and evening rush hours in the diurnal profile. This is the reason why we decided to redo OA source apportionment with the HOA reference profile as the input of the PMF-ME2 algorithm. For BBOA and CCOA, the model itself could obviously resolve these two factors without any constraint. Because the BBOA and CCOA profile may vary among different periods and different locations, we prefer not to constrain the model with BBOA and CCOA reference profile obtained from previous studies. For the cooking OA factor, we did not see any obvious signals of this factor in our solution. Therefore, we prefer not to manually force the model to resolve a cooking OA factor. The reason that a cooking OA factor is not resolved in this study is probably due to its minor concentration and contribution to total OA during wintertime in Handan, even lower than HOA.

(7) The weekdays didn't include Monday and the weekends didn't include Saturday. Was this treatment consistent with and were often used in previous studies?

Yes, this treatment was often used in previous studies. Observations indicate that Monday and Saturday behave as intermediate between weekdays and weekends due to significant memory of the previous day's emissions (Murphy et al., 2007). Therefore, many previous studies omit Monday and Saturday to reduce unnecessary bias in the weekday and weekend values, respectively (Russell et al., 2010; Revuelta et al., 2012; Choi et al., 2013; Young et al., 2016).

(8) In Figure 9, this reviewer feels the nitrate fraction increased and then decreased with the increase of PM1 loading rather than only a general decreasing trend.

We agree with the reviewer that the nitrate fraction increased a bit and then decreased with the increase of $PM_1$ loading as shown in Figure 9. The corresponding description has been revised: "The nitrate fraction went up a bit and then showed a decreasing trend with increasing $PM_1$ mass loading whereas the contribution of sulfate increased from 12% to 20% as $PM_1$ concentration developed from 100 µg/m$^3$ to 600 µg/m$^3$."

(9) Figure 11 is for the entire study period, correct? But it was placed in the case study, it probably should be made clear in the figure caption to avoid confusion.

Thanks for the suggestion. Yes, Figure 11 is for the entire study period. We put this figure in the case study to demonstrate the effects of meteorological conditions. We have made it clear in the figure caption to avoid confusion.

(10) in line 170-175, the authors stated that combustion emissions were significant as it had a good correlation with CO, but the traffic HOA in factor is only a minor contributor to the total OA. I understand the combustion does not only refer to traffic combustion emissions. But this probably should be stated clearly.

We have stated this clearly in the manuscript: "Interestingly, the temporal pattern of organics tracked well with that of CO (r = 0.84, Fig.2), implying that combustion emissions were a significant source of organic aerosols in Handan, i.e., traffic, coal combustion, and biomass burning."

**Author Responses to Anonymous Referee #2**

General Comments

This manuscript discusses a wintertime field campaign in the North China Plain during an extreme haze period. The authors evaluate the sources of primary and secondary PM and discuss the evolution of PM constituents and gaseous pollutants in light of prevailing meteorological conditions. The paper is well-written and provides full documentation of the methods, calculations, analyses performed, and conclusions based on these analyses. Given the manuscript's focus on field measurements, the evolution of pollutants within air polluted air masses, and a geographic area that hasn't been the subject of many intensive field campaigns, this manuscript falls within the scope of ACP. I have no major comments on the manuscript, but present a set of minor comments in the "Specific Comments" and "Technical Corrections" sections below. I do note that for future manuscripts, the authors should ensure that every line is numbered, as reviewers of this manuscript had to count lines to ensure that the correct line numbers were cited.

We thank the reviewer for the positive comments. We ensure that every line is numbered for future manuscripts.

Specific Comments

1. Line 44: "may cause climate change" is very vague. This should be expanded to a sentence discussing radiative forcings and the indirect and direct effects of PM.

   We have revised the sentence to "Aerosols can reduce visibility, adversely affect human health (Pope and Dockery, 2006), and influence climate change directly by absorbing and reflecting solar radiation and indirectly by modifying cloud formation and properties (Pöschl, 2005; Seinfeld and Pandis, 2012)..."

2. Line 153: The CNAAQS is quite high compared to PM standards in other countries. It would be helpful to compare the CNAAQS here to international standards to give readers a broader picture of the percent of days that had high PM.

   Thanks for the suggestion. The US National Ambient Air Quality Standards for 24-h $PM_{2.5}$ was added for comparison. The sentence was revised to "Based on TEOM measurements, only 4 days met the US National Ambient Air Quality Standards (NAAQS, 35 $\mu g/m^3$ for the 24 h average of $PM_{2.5}$) and 13 days met the Chinese NAAQS Grade II (75 $\mu g/m^3$ for the 24 h average

of PM$_{2.5}$) for the whole study period of 65 days. In other words, the daily average PM$_{2.5}$ concentrations exceeded the US NAAQS and the Chinese NAAQS on 94% and 80% of the days, respectively (Fig. 2)."

3. Line 155: Explain how "red haze alarms" are calculated.
   Because there are some criteria for the red haze alarm, a website was referenced for explanation. The sentence was revised to "On December 22, the daily PM$_{2.5}$ concentration reached the highest value of 725.7 µg/m$^3$, leading to the first "red" haze alarm (http://www.cma.gov.cn/kppd/kppdsytj/201310/t20131028_229921.html) ever in Hebei Province."

4. Figure S9: This figure is a key piece of information related to your argument in lines 170-175. I suggest moving this figure from the supplement to the main document.
   The statement in lines 170-175 is related to Figure 2 rather than Figure S9. Figure S9 only presents the diurnal pattern of the mass fractions of aerosol species. Instead of moving Figure S9, we have clearly noted "Fig.2" in the corresponding statement.

5. Line 173: I don't agree with your definition of "background." It would be more correct to simply state that ozone concentrations were nearly zero during haze episodes, as these episodes cannot be considered background time periods.
   The sentence has been revised according to the suggestion of the reviewer.

6. Line 194: Define the acronyms WS and WD here.
   Because the acronyms WS and WD have been defined in line 115, we kept these two acronyms in line 194. We have corrected the error in definition "wind direction (WS)" in line 115 to "wind direction (WD)".

7. Line 195: It's not clear how figure S10 supports this argument, and this should be clarified.
   Bivariate polar plots show how the concentration of a pollutant varies by the wind direction and wind speed at a receptor site. As shown in Figure S10, HOA demonstrated high concentrations under relatively low wind speed (<1.5m/s), suggesting that HOA was largely contributed by local emission sources rather than regional transport. For further clarification, we revised the sentence as "Bivariate polar plots, which present the concentrations of air pollutants as a function of WS and WD using the OpenAir software (Carslaw and Ropkins, 2012), demonstrated higher concentrations of HOA under relatively low WS (<1.5m/s), suggesting that HOA was substantially influenced by local emission sources, in accordance with its primary characteristics (Fig. S10)."

8. Lines 216-218: The final sentence in this paragraph is an opinion, not a result, and therefore should be moved to the conclusions.
   We think that for the section "Results and discussions", not only the results should be included but also some discussions on the results. The last sentence in line 216-218 is a discussion following the high concentration of CCOA in Handan. Therefore, we kept the sentence in this paragraph. In the conclusions, a suggestion of technology-based emission controls on coal combustion for policy makers was also mentioned.

9. Line 385: What evidence do you have that fN increased? To me, it appears that the median fN at 90-100% is the same as at 20-40%.
   Here, we were discussing the different behavior of fS and fN when RH>50%. Although the increase of fN is not significant, it showed a small increase at RH 60-70% and then decreased a bit at RH 90-100%. To avoid confusion, we revised the sentence in Line 385 to "$f_N$ showed a small increase at RH 60-70% and then decreased a bit when RH>90%".

10. Lines 412-413: Clarify what is meant by "low efficient combustion."

For clarification, we revised the sentence to "This is mainly related to large emissions of air pollutants from coal and biomass combustion during wintertime, especially for simple household stoves with low combustion efficiency."

Technical Corrections

1. Line 56: "combustions" should be "combustion"
   The text has been changed accordingly. Also changed for the title.
2. Line 225: "the" should be inserted between "during" and "biomass"
   Inserted.
3. Line 392: "For another" doesn't fit well here. I suggest changing this to "Another explanation is that"
   We have reworded accordingly.
4. Line 421: "northeastern" should be changed to "northwestern"
   Corrected.

References

Baek, B. H., Koziel, J., and Aneja, V. P.: A preliminary review of gas-to-particle conversion, monitoring, and modeling efforts in the USA. Int. J. Global Environ. Iss., 6 (2/3), 204–230, 2006.

Carslaw, D. C. and Ropkins, K.: openair - An R package for air quality data analysis, Environmental Modelling & Software, 27-500 28, 52-61, 2012.

Choi, W., Paulson, S. E., Casmassi, J., and Winer, A. M.: Evaluating meteorological comparability in air quality studies: Classification and regression trees for primary pollutants in California's South Coast Air Basin, Atmos Environ, 64, 150-159, 10.1016/j.atmosenv.2012.09.049, 2013.

Guo, S., Hu, M., Zamora, M. L., Peng, J. F., Shang, D. J., Zheng, J., Du, Z. F., Wu, Z., Shao, M., Zeng, L. M., Molina, M. J., and Zhang, R. Y.: Elucidating severe urban haze formation in China, P Natl Acad Sci USA, 111, 17373-17378, 10.1073/pnas.1419604111, 2014.

Hu, W., Hu, M., Hu, W., Jimenez, J. L., Yuan, B., Chen, W., Wang, M., Wu, Y., Chen, C., Wang, Z., Peng, J., Zeng, L., and Shao, M.: Chemical composition, sources and aging process of submicron aerosols in Beijing: contrast between summer and winter, J. Geophys. Res., 121, 1955–1977, doi:10.1002/2015JD024020, 2016.

Jiang, Q., Sun, Y. L., Wang, Z., and Yin, Y.: Aerosol composition and sources during the Chinese Spring Festival: fireworks, secondary aerosol, and holiday effects, Atmos Chem Phys, 15, 6023-6034, 10.5194/acp-15-6023-2015, 2015.

Lobert, J. M., Keene, W. C., Logan, J. A., and Yevich, R.: Global chlorine emissions from biomass burning: Reactive Chlorine Emissions Inventory, J Geophys Res-Atmos, 104, 8373-8389, Doi 10.1029/1998jd100077, 1999.

McCulloch, A., Aucott, M. L., Benkovitz, C. M., Graedel, T. E., Kleiman, G., Midgley, P. M., and Li, Y. F.: Global emissions of hydrogen chloride and chloromethane from coal combustion, incineration and industrial activities: Reactive Chlorine Emissions Inventory, J Geophys Res-Atmos, 104, 8391-8403, Doi 10.1029/1999jd900025, 1999.

Murphy, J. G., Day, D. A., Cleary, P. A., Wooldridge, P. J., Millet, D. B., Goldstein, A. H., and Cohen, R. C.: The weekend effect within and downwind of Sacramento - Part 1: Observations of ozone, nitrogen oxides, and VOC reactivity, Atmos Chem Phys, 7, 5327-5339, 2007.

Pope III, C. A. and Dockery, D. W.: Health Effects of Fine Particulate Air Pollution: Lines that Connect, J. Air Waste Manage. 56, 709–742, 2006.

Pöschl, U.: Atmospheric Aerosols: Composition, Transformation, Climate and Health Effects, Angew. Chem. Int. Ed., 44, 7520–7540, 2005.

Revuelta, M. A., Harrison, R. M., Nunez, L., Gomez-Moreno, F. J., Pujadas, M., and Artinano, B.: Comparison of temporal features of sulphate and nitrate at urban and rural sites in Spain and the UK, Atmos Environ, 60, 383-391, 10.1016/j.atmosenv.2012.07.004, 2012.

Russell, A. R., Valin, L. C., Bucsela, E. J., Wenig, M. O., and Cohen, R. C.: Space-based Constraints on Spatial and Temporal Patterns of NOx Emissions in California, 2005-2008, Environ Sci Technol, 44, 3608-3615, 10.1021/es903451j, 2010.

Seinfeld, J. H. and Pandis, S. N.: Atmospheric Chemistry and Physics: From Air Pollution to Climate Change, John Wiley & Sons, New York, 2nd edition, 1232 pp., ISBN-13: 978-0-471-72018-8, 2006.

Sun, Y. L., Wang, Z. F., Fu, P. Q., Yang, T., Jiang, Q., Dong, H. B., Li, J., and Jia, J. J.: Aerosol composition, sources and processes during wintertime in Beijing, China, Atmos. Chem. Phys., 13, 4577-4592, doi:10.5194/acp-13-4577-2013, 2013.

Sun, Y., Jiang, Q., Wang, Z., Fu, P., Li, J., Yang, T., and Yin, Y.: Investigation of the Sources and Evolution Processes of Severe Haze Pollution in Beijing in January 2013, J. Geophys. Res., 119, 4380–4398, 2014.

Sun, Y. L., Wang, Z. F., Du, W., Zhang, Q., Wang, Q. Q., Fu, P. Q., Pan, X. L., Li, J., Jayne, J., and Worsnop, D. R.: Long-term real-time measurements of aerosol particle composition in Beijing, China: seasonal variations, meteorological effects, and source analysis, Atmos. Chem. Phys., 15, 10149–10165, doi:10.5194/acp-15-10149-2015, 2015.

Young, D. E., Kim, H., Parworth, C., Zhou, S., Zhang, X. L., Cappa, C. D., Seco, R., Kim, S., and Zhang, Q.: Influences of emission sources and meteorology on aerosol chemistry in a polluted urban environment: results from DISCOVER-AQ California, Atmos Chem Phys, 16, 5427-5451, 10.5194/acp-16-5427-2016, 2016.

Zhang, J. K., Sun, Y., Liu, Z. R., Ji, D. S., Hu, B., Liu, Q., and Wang, Y. S.: Characterization of submicron aerosols during a month of serious pollution in Beijing, 2013, Atmos. Chem. Phys., 14, 2887-2903, doi:10.5194/acp-14-2887-2014, 2014.